# Relationship between the Mediterranean Diet and Metabolic Syndrome and Each of the Components That Form It in Caucasian Subjects: A Cross-Sectional Trial

**DOI:** 10.3390/nu16121948

**Published:** 2024-06-19

**Authors:** Leticia Gómez-Sánchez, Marta Gómez-Sánchez, Olaya Tamayo-Morales, Cristina Lugones-Sánchez, Susana González-Sánchez, Ruth Martí-Lluch, Emiliano Rodríguez-Sánchez, Luis García-Ortiz, Manuel A. Gómez-Marcos

**Affiliations:** 1Emergency Service, University Hospital of La Paz P. of Castellana, 261, 28046 Madrid, Spain; 2Home Hospitalization Service, Marqués of Valdecilla University Hospital, s/n, 39008 Santander, Spain; martagmzsnchz@gmail.com; 3Primary Care Research Unit of Salamanca (APISAL), Salamanca Primary Care Management, Institute of Biomedical Research of Salamanca (IBSAL), 37005 Salamanca, Spain; olayatm@usal.es (O.T.-M.); cristinals@usal.es (C.L.-S.); gongar04@gmail.com (S.G.-S.); emiliano@usal.es (E.R.-S.); lgarciao@usal.es (L.G.-O.); 4Castilla and León Health Service-SACYL, Regional Health Management, 37005 Salamanca, Spain; 5Red de Investigación en Cronicidad, Atención Primaria y Promoción de la Salud (RICAPPS), 37005 Salamanca, Spain; rmarti.girona.ics@gencat.cat; 6Vascular Health Research Group, Instituto Universitario para la Investigación en Atención Primaria de Salud Jordi Gol i Gurina (IDIAPJGol), 08007 Girona, Spain; 7Girona Biomedical Research Institute (IDIBGI), Doctor Trueta University Hospital, 17190 Girona, Spain; 8Department of Medical Science, Faculty of Medicine, University of Girona (UdG), 17003 Girona, Spain; 9Department of Medicine, University of Salamanca, 28046 Salamanca, Spain; 10Department of Biomedical and Diagnostic Sciences, University of Salamanca, 37007 Salamanca, Spain

**Keywords:** Mediterranean diet, metabolic syndrome, Caucasian subjects, cross-sectional trial

## Abstract

The main objective of this work is to investigate the relationship between the Mediterranean diet (MD) and metabolic syndrome (MetS) and its components in Caucasian subjects between 35 and 74 years. The secondary objective is to analyze sex differences. Methods: A cross-sectional trial. This study utilized data from the EVA, MARK, and EVIDENT studies, and a total of 3417 subjects with a mean age ± SD of 60.14 ± 9.14 years (57% men) were included. We followed the five criteria established in the National Cholesterol Education Program III to define MetS. The MD was assessed with the 14-item Mediterranean diet adherence screener (MEDAS) used in the PREDIMED study. Good adherence was considered when the MD value was higher than the median value. Results: The mean ± SD value of the MEDAS questionnaire was 5.83 ± 2.04 (men 5.66 ± 2.06 and women 6.04 ± 1.99; *p* < 0.001). Adherence to the MD was observed by 38.6% (34.3% men and 40.3% women; *p* < 0.001). MetS was observed in 41.6% (39.0% men and 45.2% women; *p* < 0.001). In the multiple regression analysis, after adjusting for possible confounders, the mean MD value showed a negative association with the number of MetS components per subject (β = −0.336), and with the different components of MetS: systolic blood pressure (β = −0.011), diastolic blood pressure (β = −0.029), glycemia (β = −0.009), triglycerides (β = −0.004), and waist circumference (β = −0.026), except with the HDL-cholesterol value which showed a positive association (β = 0.021); *p* < 0.001 in all cases. In the logistic regression analysis performed, we found that an increase in MD adherence was associated with a decrease in the probability of MetS (OR = 0.56) and its components: blood pressure levels ≥ 130/85 mmHg (OR = 0.63), fasting plasma glucose ≥ 100 mg/dL (OR = 0.62), triglyceride levels ≥ 150 mg/dL (OR = 0.65), waist circumference levels ≥ 88 cm in women and ≥102 cm in men (OR = 0.74), and increased high-density lipoprotein cholesterol < 40 mg/dL in men and <50 mg/dL in women (OR = 1.70); *p* < 0.001 in all cases. The results by sex were similar, both in multiple regression and logistic regression. Conclusions: The results found in our work indicate that the greater the adherence to the MD, the lower the probability of presenting MetS. This result is repeated in the study by sex. More studies are needed to clarify that these results can be extended to the rest of the Mediterranean countries, and to other countries outside the Mediterranean basin.

## 1. Introduction

Nutritional research currently focuses on dietary patterns rather than individual nutrients or foods [1]. The Mediterranean diet (MD) is one of the dietary patterns on which the most studies have been carried out in the world. Its main characteristics are based on the consumption of a wide variety of foods such as: extra virgin olive oil, all types of legumes, cereals, nuts such as walnuts, fruits and vegetables, low-fat dairy products, fresh fish, and red wine [2], and there is increasing evidence that dietary patterns inspired by the MD are associated with multiple health benefits [3,4]. Many of these foods offer an array of phytonutrients, with polyphenols and vitamins being particularly significant. This dietary pattern is considered one of the healthiest, primarily because of its antioxidant and anti-inflammatory properties [1,5,6]. Numerous studies have demonstrated that adhering to a MD is a crucial factor in promoting a healthy lifestyle and preventing various chronic diseases [4]. This is demonstrated by a systematic review that found inverse associations between greater MD adherence and overweight/obesity and chronic diseases associated with obesity [7]. A meta-analysis of 4 randomized controlled trials and 32 cohort studies found that higher adherence to the MD was associated with improved cardiovascular health outcomes, including significant reductions in rates of cardiac ischemia, cerebrovascular ischemia, and total heart diseases [8]. Likewise, numerous studies have confirmed the protective effects of the MD against various chronic diseases, including type 2 diabetes mellitus [9,10,11], cardiovascular diseases [12,13], cognitive impairment [14], certain types of cancer [15], aging disorders [16,17], and metabolic syndrome (MetS) [18]. Likewise, greater adherence to the MD was linked to a 23% reduction in the risk of all-cause mortality [19]. All of this is related to a better lipid profile, lower blood pressure (BP), lower blood sugar, and reduced rates of obesity [6]. 

MetS consists of various risk factors for atherosclerosis, such as obesity, high blood pressure, elevated fasting plasma glucose (FPG), and atherogenic dyslipidemia [20]. MetS has emerged as a significant global health issue. Its prevalence is rising due to an increase in the consumption of fast food (high in calories and low in fiber) and a decrease in physical activity resulting from mechanized transport and sedentary activities during leisure time [21]. MetS increases the likelihood of developing cardiovascular disease morbidity and mortality twofold, while also raising the risk of all-cause mortality by 50% [22,23,24]. 

Healthy lifestyle factors, including physical activity and adherence to the MD, are associated with reduced incidence of MetS. A recent study indicated that individuals with higher MetS scores often follow a dietary pattern that is pro-inflammatory and exhibit lower adherence to the MD. This includes lower consumption of carbohydrates and nuts, and higher intake of proteins, saturated fatty acids, cholesterol, red and processed meats, and oils other than olive oil [25].

The study of the Mediterranean diet in recent years has focused on the study of adherence to the MD in populations from different geographical areas, according to age, activity, profession, country of origin, difficulties in carrying it out, and its relationship with certain pathologies, including MetS [1]. However, there are fewer studies that analyze the relationship between the MD with SD and with each of its components, as we propose in this work. Thus, a sub-analysis carried out with data from the PREDIMED-Plus randomized controlled trial assessed the effects of an intervention aimed at losing weight based on a MD with caloric restrictions and the promotion of physical activity, compared to MD recommendations without caloric restrictions, with the conclusion that the MD combined with caloric restrictions plus increased physical activity reduced plasma triglycerides and increased HDL-cholesterol particles, and also reduced weight, indicating beneficial changes against cardiovascular disease [18]. Furthermore, a review suggests that MD adherence can be considered the first step in the treatment of MetS [26]. 

However, the impact of the Mediterranean diet (MD) on the prevalence of this disease and its individual components in the Caucasian population remains incompletely understood. Therefore, the primary objectives of this study are twofold: firstly, to examine the association between the MD and metabolic syndrome (MetS) as well as its components among Caucasian individuals aged 35 to 74 years; secondly, to analyze potential differences between sexes.

## 2. Materials and Methods

### 2.1. Design

A descriptive cross-sectional study was conducted with data from the EVA study [27] (NCT02623894), the MARK study [28] (NCT01428934), and EVIDENT study [29] (NCT0108308). 

### 2.2. Study Population

All studies were conducted in primary care, involving 4229 subjects between the ages of 35 and 75. A total of 3417 subjects were included in this work. A total of 812 subjects were excluded from this work as they had not registered all the variables necessary to carry out this work. Specifically, the following were included: 491 of the 501 subjects recruited from the EVA project [27], selected by random sampling from the urban population without cardiovascular disease (reference population 43,946); 2120 of the 2511 subjects included from the MARK project [28], selected by random sampling among the subjects consulted in 6 urban health centers and with inert cardiovascular risk; and 806 of the 1217 subjects from the EVIDENT project [29], selected by random sampling among the subjects consulted in a primary care center. The flowchart of the subjects included and excluded from each of the three projects are presented in Figure 1.

The MARK project [28] is a prospective multicenter cohort study. The primary aim of the study was to determine the ability of different parameters such as the ankle-brachial index, CAVI, and postprandial glycemia in the prediction of cardiovascular risk in subjects with intermediate cardiovascular risk. Subjects from 6 health centers distributed in 3 autonomous communities, who presented intermediate cardiovascular risk were included, in total numbering 2511. Cardiovascular risk was defined by the Framingham scale adapted to the Spanish population [30] and by the systematic coronary risk evaluation (SCORE) scale [31] The mean age observed was 61 ± 8 years, 62% of whom were men. The determinants of accelerated vascular aging (EVA) project [27] is a prospective multicenter cohort project. The primary aims of the study were: to establish reference values for vascular structure and function parameters within the urban population, analyze the influence of modifiable factors on vascular ageing, and to analyze the different criteria used in the definition of vascular ageing. A total of 511 subjects from 4 health centers in the urban population of Salamanca, without previous cardiovascular disease, were included. The mean age observed was 56 ± 14 years, 50% of whom were men. The EVIDENT study [29] is a clinical trial conducted using a randomized, double-blind, parallel-group design. The primary aims of the study were: in the first phase, to develop and validate a smartphone application, and then to assess the impact of using this technology in comparison with a standardized intervention aimed at improving adherence to MD nutritional recommendations and increasing physical activity. The study involved seven groups from seven autonomous communities. The mean age observed was 52 ± 12 years, and 48% were men. A comprehensive outline of the study methodologies, including the criteria for inclusion and exclusion, can be found in the protocols of the three studies [27,28,29].

### 2.3. Variables and Measuring Instruments

#### 2.3.1. Adherence to the Mediterranean Diet 

Adherence to the MD was assessed using a 14-item questionnaire, which was validated in Spain and employed in the PREDIMED study [32]. The questionnaire comprises 12 inquiries regarding the frequency of food consumption and 2 queries about typical eating habits observed in the Spanish population. Each question was assigned either zero or one point. Points were awarded for: using olive oil as the primary fat for cooking, consuming four or more tablespoons (one tablespoon = 13.5 g) of olive oil daily (including for frying and salad dressing), having two or more servings of vegetables, consuming three or more pieces of fruit, consuming less than one serving of red or processed meat, consuming less than one serving of animal fat, drinking less than one cup (one cup = 100 mL) of sugary carbonated beverages, preferring white meat over red meat, consuming seven or more glasses of wine weekly, consuming three or more servings of legumes, consuming three or more servings of fish, consuming three or more servings of nuts or dried fruit, consuming two or more servings of sofrito (a traditional sauce made with tomato, garlic, onion, or leeks sautéed with olive oil), and consuming less than two servings of baked goods. The total score ranged from 0 to 14 points, with scores above the median (7) indicating adherence to the MD [32].

#### 2.3.2. Diagnostic Criteria of Metabolic Syndrome

Following the recommendations of the international consensus of the National Cholesterol Panel Education Program Adult Treatment [20], MetS was defined as subjects who had 3 or more of the 5 components shown in Table 1. 

#### 2.3.3. Anthropometric Measurements and Cardiovascular Risk Factors

Body weight was assessed twice using a calibrated electronic scale (Seca 770; medical scale and measurement systems, Birmingham, UK) with a precision of ±0.1 kg. The measurements were taken with the patient wearing light clothing and without shoes and rounded to the nearest 100 g. Height was measured using a portable system (Seca 222; medical scale and measurement systems, Birmingham, UK), averaging two readings with the patient shoeless and standing. Height values were rounded to the nearest centimeter. Body mass index (BMI) was calculated as weight in kilograms divided by height squared in meters (kg/m^2^). Waist circumference was measured following the 2007 guidelines of the Spanish Society for the Study of Obesity [33].

Office blood pressure was measured using a validated OMRON model M10-IT sphygmomanometer (Omron Health Care, Kyoto, Japan), following the guidelines of the European Society of Hypertension. Three measurements of systolic blood pressure (SBP) and diastolic blood pressure (DBP) were taken, and the average of the last two readings was recorded [34]. Subjects were classified as follows: hypertension if they were prescribed antihypertensive medication or had blood pressure readings ≥ 140/90 mmHg; diabetes if they were taking hypoglycemic agents or had fasting plasma glucose levels ≥ 126 mg/dL or HbA1c ≥ 6.5%; dyslipidemia if they were on lipid-lowering medications or had fasting total cholesterol levels ≥ 240 mg/dL, low-density lipoprotein cholesterol (LDL-C) ≥ 160 mg/dL, high-density lipoprotein cholesterol (HDL-C) < 40 mg/dL for men and <50 mg/dL for women, or triglycerides ≥ 150 mg/dL. Obesity was defined as a BMI ≥ 30 kg/m^2^ [35]

### 2.4. Statistical Analysis

The data were reported using means ± standard deviations for continuous variables and numbers or percentages for categorical variables. Statistical comparisons between men and women were conducted using chi-square tests for percentages and Student’s *t*-tests for continuous variables. Pearson correlation analysis was employed to investigate relationships between continuous variables.

To analyze the association between average score value of the MD and the number of MetS components, seven multiple linear regression models were conducted using the ENTER method, with the MD as the independent variable and the number of MetS components, SBP in mmHg, DBP in mmHg, FPG in mg/dL, triglycerides in mg/dL, HDL cholesterol in mg/dL, and WC in cm, as dependent variables.

To analyze the association between adherence to the MD and the presence of MetS and each of its components, we used six logistic regression models. MD adherence was the independent variable (encoded as MD adherence = 1, MD non-adherence = 0). MetS components were the dependent variables (encoded as yes = 1, no = 0), BP ≥ 130/85 mmHg (yes = 1, no = 0), FPG ≥ 100 mg/dL (yes = 1, no = 0), TGC ≥ 150 mg/dL (yes = 1, no = 0), HDL-C mg/dL < 40 in men, <50 mg/dL in women (yes = 1, no = 0), and WC ≥ 88 cm in women, ≥102 cm in men (yes = 1, no = 0). All models included the adjustment variables of age, sex, and the consumption of antihypertensive drugs, hypoglycemic agents, and lipid-lowering agents.

All analyses were performed globally and by sex. The SPSS statistics program for Windows, version 28.0 (IBM Corp, Armonk, NY, USA) was used. We considered a value of *p* < 0.05 as a statistical significance limit.

### 2.5. Ethical Principles

The research protocols for the studies incorporated into this pooled analysis were duly approved by the Drug Research Ethics Committee of Salamanca, with registration numbers PI15/01039 and PI20/10569 (EVA study [27]), PI10/02043 (MARK study [28]), and PI83/06/2018 (EVIDENT study [29]). All subjects participating in these studies provided written informed consent. During the development of the study, the standards of the Declaration of Helsinki [36] and the WHO standards for observational studies were followed. The confidentiality of the subjects included was always guaranteed in accordance with the provisions of Organic Law 3/2018, of 5 December, on Personal Data Protection and Guarantee of Digital Rights and Regulation (EU) 2016/679 of the European Parliament and of the Council of 27 April 2016 on Data Protection (RGPD).

## 3. Results

### 3.1. General Characteristics of the Subjects Analyzed 

The characteristics, cardiovascular risk factors, and drug use among the total number of subjects and in each of the sexes are described in Table 1. The mean age was 60.14 ± 9.77 years, with no differences observed between sexes (*p* = 0.543). Men made up a greater proportion of the sample (57%), and showed higher BP, LDL cholesterol, TGC, FPG, and WC values than women. Women had higher total cholesterol and HDL-cholesterol values than men.

### 3.2. MD and Metabolic Syndrome

Table 2 shows the values for MD and MetS. The mean value of the total MD score was 5.83 ± 2.04 (women 6.04 ± 1.99, men 5.67 ± 2.06). The adherence to MD percentage was 36.8% (women 40.3%, men 34.3%). The percentage of subjects with MetS was 41.6% (women 45.2%, men 39.0%) and the mean number of MetS components was 2.28 ± 1.33 (women 2.33 ± 1.44, men 2.24 ± 1.25). The mean number of MetS components in subjects with MetS was 3.60 ± 0.73 (women 3.66 ± 0.75, men 3.54 ± 0.70). When analyzing each of the components, women had a higher percentage of WC (69.3 vs. 48.6) and HDL-C (37.2 vs. 21.9), and a lower percentage of FPG (34.2 vs. 40.6), BP (68.0 vs. 80.2), and TGC (24.1 vs. 32.6) than men. 

Figure 2 shows the overall distribution of subjects according to whether they have MetS or adherence to MD, and Figure 3 shows the distribution by sex.

Figure 4 shows the difference in the mean value of the MD score between subjects with and without MetS and without each of the components of MetS and with MetS overall and by sex. In MetS, the FPG, HDL-C, and WC components were greater in women, and in the TGC the difference was greater in men (*p* < 0.05).

### 3.3. Correlation between the MD and MetS and Its Components

The mean value of MD showed a negative correlation with the number of MetS components r = −0.196 (women r = −0.220; men r = −0.185), and with all MetS components in the overall analysis and by sex, except for HDL-C, which showed a positive correlation r = 0.171 (women r = 0.179; men r = 0.130) (Table 3).

### 3.4. Association between Adherence to the MD and MetS and Its Components

In the multiple regression analysis, adjustments were made for potential confounders, the mean value of MD showed a negative association with the number of MetS components (β = −0.336, 95% CI −0.393 to −0.280), and with the different MetS components: SBP (β = −0.011, 95% CI −0.015 to −0.008), DBP (β = −0.029, 95% CI −0.035 to −0.022), FPG (β = −0.009, 95% CI −0.012 to −0.006), TGC (β = −0.004, 95% CI −0.004 to −0.003), and WC (β = −0.026, 95% CI −0.032 to −0.020), except with the HDL-cholesterol value which showed a positive association (β = 0.021, 95% CI 0.018 to 0.028). The results by sex were similar (Table 4). 

Table 5 shows the results of the logistic regression analysis overall and by sex. The adjusted logistic regression models indicated that an increase in MD adherence decreases the probability of MetS (OR = 0.555, 95% CI 0.477 to 0.650) and its components: BP ≥ 130/85 mmHg (OR = 0.634, 95% CI 0.522 to 0.771), FPG ≥ 100 mg/dL (OR = 0.615, 95% CI 0.511 to 0.740); TGC ≥ 150 mg/dL (OR = 0.641, 95% CI 0.546 to 0.753); WC ≥ 88 cm in women and ≥102 cm in men (OR = 0.741, 95% CI 0.639 to 0.859), and increases HDL cholesterol < 40 mg/dL in men and <50 mg/dL in women (OR = 1.700), 95% CI 1.442 to 2.005 (*p* < 0.001 in all cases). The results by sex were similar. 

## 4. Discussion

The findings from the study involving 3417 Caucasian subjects indicate that approximately 4 out of 6 individuals have MetS, with a higher percentage observed among men. Similarly, MD adherence was 37%, with 34% in men and 40% in women. The percentage of each of the MetS components varied according to sex. The presence of MetS and all components showed an association with the MD overall and by sex.

The present study found that the percentage of men with MetS was higher than that of women. These results are consistent with those reported in the Darios study [37] and partially so with those published in the ENRICA study [38], which showed that while the percentage of men with MetS was higher than women up to 65 years, the trend is inverted thereafter. On the other hand, the MD adherence percentage was higher in women than in men, as shown in the research published in 2020 comparing MD adherence with vascular aging [39]. These results do not overlap with all published research, for example in the work conducted by Minji Kang et al. [40], which included 160,353 participants from different countries to assess diet quality. They found that in different countries, adherence to the MD was higher in men than in women. The discrepancies with our findings may potentially be explained by differences in the age distribution of the population studied, their country of origin, and the distribution of percentages by sex.

The prevalence of MetS in this manuscript was 41% (45% in women and 39% in men) and is higher than that reported in other studies conducted in Spain, which found the overall prevalence of MetS to be lower and, in the analysis by sex, higher in women than in men. Thus, in the ENRICA study [38], which included 11,149 representatives of the Spanish population aged over 18 years, the prevalence in the 45–64 age group was 31% (36% men and 25% women) and increased with age and in the communities of southern Spain. In the DARIOS study [37], which included 24,670 people aged between 35 and 74 years from 10 autonomous communities, the prevalence of MetS was 31% (32% in men and 29% in women). In the same vein, the prevalence of MetS was higher than in the United States, being 34% (36% in men and 32% in women) [41]. The higher prevalence of MetS in this research can be explained thus: the subjects included in the MARK study (2120 subjects; representing 61% of the total), had intermediate cardiovascular risk and the prevalence of MetS in this group was 52% (61% in women and 45% in men), which would also explain the discrepancies between sexes. Conversely, in the subjects from the EVA study (491 subjects; representing 14% of the total), coming from a general population sample without previous cardiovascular disease, the prevalence of MetS was 14% (13% in women and 15% in men).

When analyzing each of the components, as well as the results found in this study, the percentage of FPG ≥ 100 mg/dL and TGC ≥ 150 mg/dL was higher in men. The percentage of subjects with HDL-C < 40 mg/dL in men, <50 mg/dL in women, and WC ≥ 88 cm in women, ≥102 cm in men, was higher in women. There is considerable variation in prevalence depending on the geographical area, age, sex, level of education, degree of obesity, and the definition used for diagnosis [37,38,42]. However, we must bear in mind that these results are not comparable since they correspond to different age groups, and subjects from the different studies had different characteristics. In our case, a significant percentage of those from the MARK study [28] were subjects with intermediate cardiovascular risk, while those from the EVIDENT [29] study were selected by random sampling among the subjects consulted, and subjects from a population-based sample were only included in the EVA study [27]. Moreover, while the ENRICA study [38] included subjects from all over Spain over 18 years of age, the DARIOS study [37] only included data from 10 autonomous communities and from subjects aged between 35–75 years. 

However, there are fewer data in longitudinal studies on the effect of the MD on the incidence of MetS. Some longitudinal studies have examined the importance of Mediterranean lifestyles, as measured with the adherence to the Mediterranean lifestyle (MEDLIFE) questionnaire. A five year follow up of the subjects included in the CORDIOPREV study showed that those with greater adherence to the Mediterranean lifestyle had a lower incidence of MetS (odds ratio 0.37; 95% CI: 0.19–0.75) and a higher likelihood of reversing MetS (odds ratio 2.08; 95% CI: 1.11–3.91) compared to participants in the low adherence group of MEDLIFE [43]. Similarly, with the cohort of the ENRICA study at 8.7 years of follow up, assessing the Mediterranean lifestyle with the MEDLIFE index demonstrated that higher adherence to it was associated with a lower incidence of MetS [44]. The MEDLIFE index, representing the Mediterranean lifestyle, encompasses not only food consumption but also other dietary habits and healthy behaviors such as communal living, eating together, rest, and social habits typical of traditional Mediterranean culture. This underscores the significance of cultural practices beyond mere dietary choices as influential factors in health outcomes. Previous studies have examined the combined and potentially synergistic effects of these behaviors related to traditional Mediterranean culture, rather than isolating the effects of each item individually. Therefore, prospective studies are warranted to specifically analyze the impact of adherence to the Mediterranean diet on MetS.

In this study, we found an association between the variables that define the MD and MetS as continuous variables and categorical variables. This association differs from that found in other studies, for example, a study conducted on 1404 adults in Luxembourg only found a significant association between the MD and MetS when used as a continuous score, based on the weighting of the compounds by exploratory factor analysis with MetS, but not when used as categorical variables [45]. However, the PREDIMED-PLUS study that included 5739 overweight/obese participants with MetS characteristics (aged 55 to 75 years) showed that participants with MetS tended to have lower adherence to the MD [25]. In the study carried out by Hassani, S et al. [46], adherence to the MD was not associated with MetS, presenting an association only with fasting blood glucose (OR: 0.57, CI: 0.33–0.97) and abdominal obesity (OR: 0.42, CI: 0.20–0.87) in women. In 1972, Greek schoolchildren aged 9 to 13 years with a logistic regression analysis revealed that “poor” adherence to the MD was associated with an increased likelihood of central obesity (OR 1.31; 95% CI: 1.01–1.73) and hypertriglyceridemia (OR 2.80; 95% CI: 1.05–7.46), after adjusting for several possible confounders [47]. Finally, a meta-analysis that included 58 studies and analyzed the relationship between adherence to the MD and the components of MetS found that WC β = −0.20, (95% CI: −0.40, −0.01) and TGC β = −0.27 (95% CI: −0.27, −0.11) were lower, and HDL-C β = −0.28 (95% CI: 0.07, 0.50) was higher in the group with high adherence to the MD without finding an association with FBG and BP [48].

In summary, high adherence to the MD can have a positive impact on MetS parameters. However, this may vary from one component to another, probably explained by the heterogeneity in the studies in terms of subjects included, analyses performed, associated pathologies, or concomitant treatments. More research is, therefore, needed in this field.

Additionally, while our results derive from a representative Spanish population, it is important to investigate their generalizability to non-Mediterranean countries or different ethnic groups. Although the MD aims to encompass a comprehensive Mediterranean lifestyle index, other factors such as outdoor activities and levels of physical activity may also influence these results [25]. Moreover, other factors may be involved. A recent study, using artificial intelligence analysis, related adherence to the Mediterranean diet, BP, HDL levels, triglycerides, basal glycemia, and MetS in males and females to their olfactory function. This is not an aspect to be underestimated given the role that smell plays in eating behavior and food choice [49]. For all these reasons, the main novelty of this study is that, as far as we know, it is the first to analyze the impact of adherence to the MD on MetS and on each of its components in the Spanish population using validated scores of adherence to the MD to assess the level of adherence to the MD [16].

### Limitations and Strengths

This study has several limitations and strengths. The primary limitation is the analysis of cross-sectional data, which restricts our ability to establish causality. Another limitation is that the people included in the analysis come from three studies with very different characteristics, including the largest number of subjects in the MARK study. Among the strengths of the study are the sample size and the rigorous adherence to standardized conditions for all criteria defining MetS, including anthropometry and blood pressure measurements conducted with validated devices. In addition, all analytical measurements were performed in laboratories with adequate quality controls.

## 5. Conclusions

The results of this work indicate that higher adherence to the MD significantly decreases the likelihood of developing MetS and its individual components, both overall and when analyzed by sex. These findings underscore the importance of promoting adherence to the MD within primary health care settings to reduce the prevalence of MetS and its associated health complications. Future research should explore whether these benefits are consistent in other Mediterranean countries and assess the potential for improving cardiovascular health in populations outside the Mediterranean basin. Additionally, it remains uncertain whether the MD has the same beneficial effects in both healthy and diseased populations, highlighting the need for further studies in this area.

## Figures and Tables

**Figure 1 nutrients-16-01948-f001:**
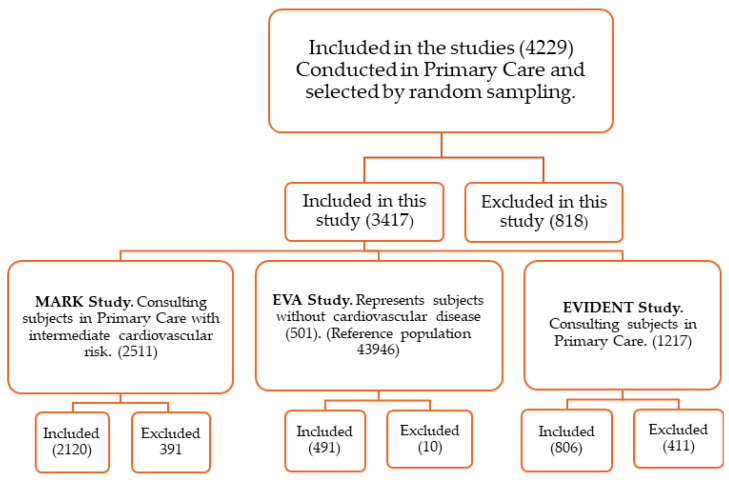
This flowchart describes the provenance of the 3417 subjects included in the selection by random sampling of the subjects in each study, up to the combination of the data and the final analysis.

**Figure 2 nutrients-16-01948-f002:**
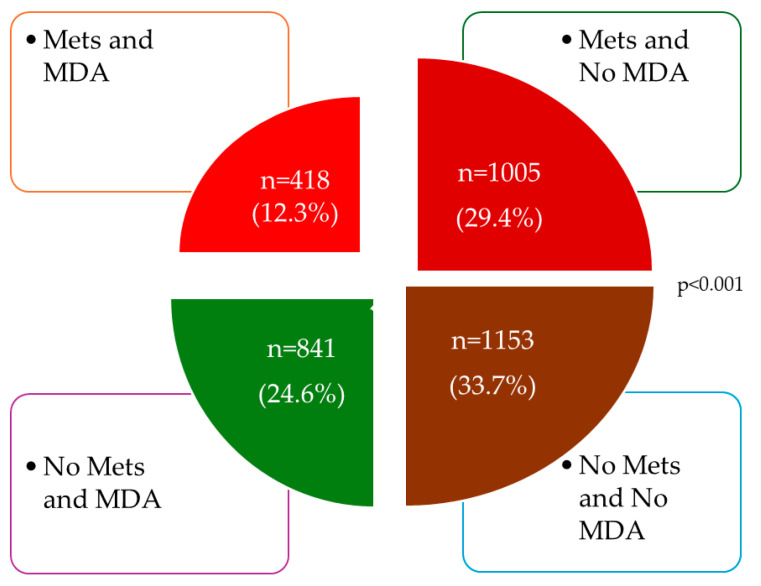
Distribution of subjects according to whether they have MetS or adherence to MD. MDA, Mediterranean diet adherence; MetS, metabolic syndrome.

**Figure 3 nutrients-16-01948-f003:**
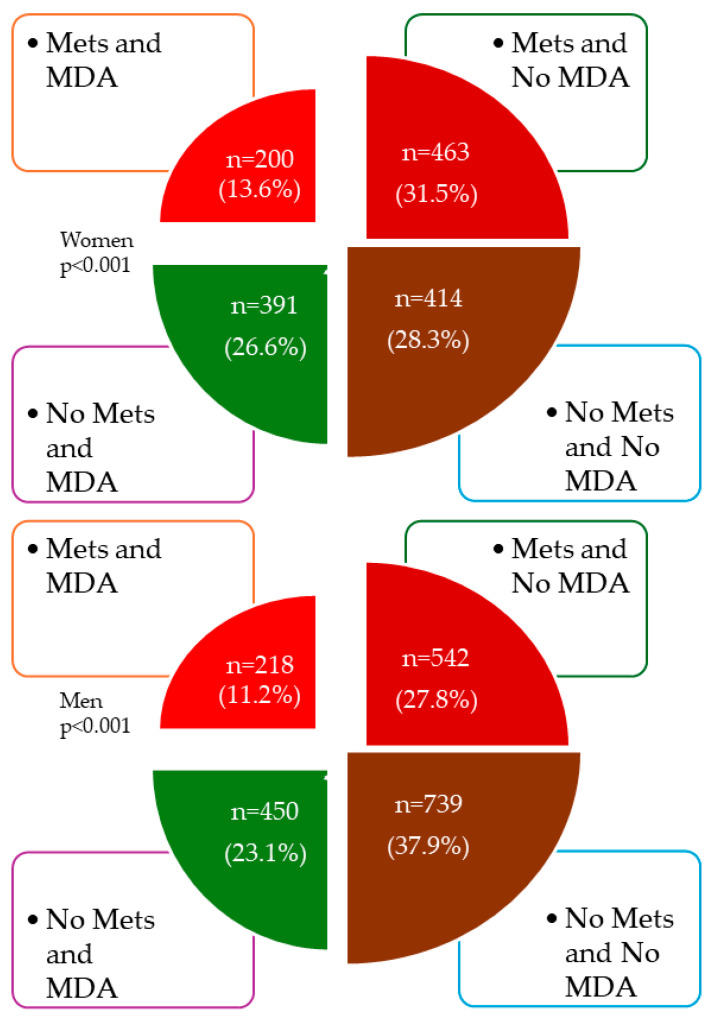
Distribution of subjects according to whether they have MetS or adherence to MD by sex. MDA, Mediterranean diet adherence; MetS, metabolic syndrome.

**Figure 4 nutrients-16-01948-f004:**
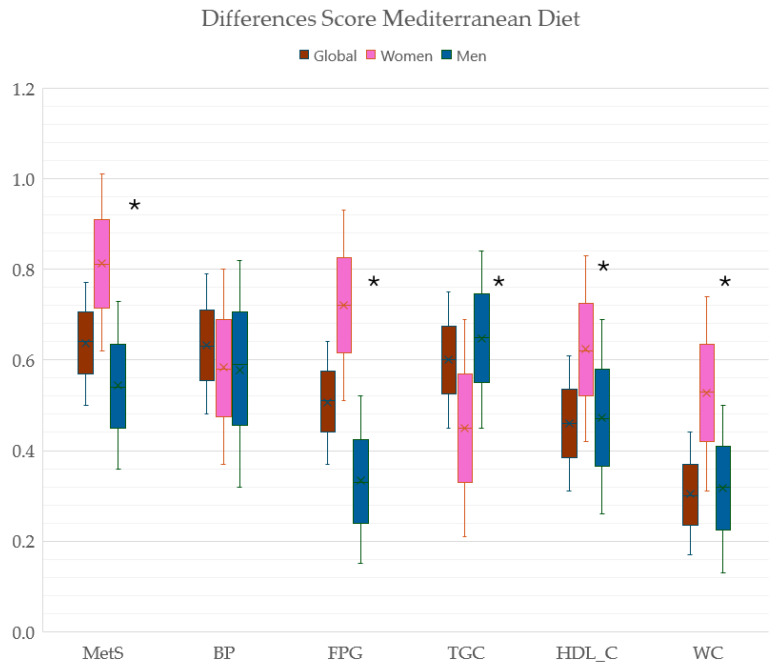
Differences and 95% CI in the mean value of the MD score and its components between subjects with and without MetS and its components. MD, Mediterranean diet; MetS, metabolic syndrome; BP, blood pressure ≥ 130/85 mmHg; FPG, fasting plasma glucose ≥ 100 mg/dL, TGC, triglycerides ≥ 150 mg/dL; HDL-C, high–density lipoprotein < 40 in men mg/dL, <50 mg/dL in women; WC, waist circumference ≥ 88 cm in women, ≥102 cm in men. * *p* < 0.005 between men and women.

**Table 1 nutrients-16-01948-t001:** Diagnostic Criteria for MetS.

Abdominal obesity	Waist circumference ≥ 88 cm in women and ≥102 cm in men
Triglycerides	≥150 mg/dL or lipid-lowering therapy
HDL Cholesterol	<40 mg/dL in men or <50 mg/dL in women
Glycemia	Fasting blood glucose ≥ 100 mg/dL or treatment with hypoglycemic agents
Blood pressure	SBP ≥ 130 mm Hg or DBP ≥ 85 mmHg or treatment with antihypertensives

HDL, high-density lipoprotein; SBP, systolic blood pressure; DBP, diastolic blood pressure. Adapted from NCPIII [32].

**Table 2 nutrients-16-01948-t002:** General characteristics of the subjects included globally and by sex.

	Total (n = 3417)		Women (n = 1468)		Men (n = 1849)		*p* Value
**MD**							
MD (total score)	5.83	±2.04	6.04	±1.99	5.67	±2.06	<0.001
Adherence to MD, n (%)	1259	(36.8)	509	(40.3)	668	(34.3)	<0.001
**Conventional** **risk factors**							
Sex, n (%)	-----		1468	(43.0)	1849	(57.0)	<0.001
Age, (years)	60.14	±9.77	60.02	±10.02	60.23	9.58	0.543
SBP, (mmHg)	133.32	±19.37	128.67	±20.67	136.83	17.53	<0.001
DBP, (mmHg)	81.93	±10.94	79.54	±10.81	83.74	10.69	<0.001
Hypertension, n (%)	2200	(64.4)	855	(58.2)	1345	(69.0)	<0.001
Antihypertensive drugs, n (%)	1564	(45.8)	654	(44.6)	910	(46.7)	0.214
Total cholesterol, (mg/dL)	216.07	±41.28	219.90	±42.74	213.18	±39.91	<0.001
LDL cholesterol, (mg/dL)	132.74	±35.23	131.99	±35.95	133.31	±34.67	0.139
HDL cholesterol, (mg/dL)	52.57	±14.51	57.30	±15.79	49.00	±12.32	<0.001
Triglycerides, (mg/dL)	132.94	±83.62	122.17	±68.49	141.05	±92.61	<0.001
Dyslipidemia, n (%)	2721	(79.7)	1182	(80.5)	1539	(79.0)	0.148
Lipid–lowering drugs, n (%)	976	(28.6)	433	(29.5)	543	(27.9)	0.156
FPG, (mg/dL)	101.60	±31.72	100.08	32.27	102.75	31.25	0.007
HbA1c, (%)	5.94	±1.05	5.93	±1.07	5.95	±1.04	0.680
Diabetes mellitus, n (%)	687	(20.1)	282	(19.2)	405	(20.8)	0.138
Hypoglycemic drugs, n (%)	575	(16.8)	235	(16.0)	340	(17.4)	0.143
Weight, kg	77.44	±14.66	70.17	±13.34	82.92	±13.14	<0.001
Height, cm	164.54	±9.46	157.09	±6.68	170.15	±7.08	<0.001
BMI, (kg/m^2^)	28.55	±4.54	28.47	±5.31	28.60	±3.86	0.202
WC, cm	98.58	±12.12	94.28	±12.94	101.82	±10.35	<0.001
Obesity, n (%)	1079	(31.6)	469	(33.8)	583	(39.0)	<0.001
**MetS and its components**							
Number of MetS components	2.28	±1.33	2.33	±1.44	2.24	±1.25	0.027
MetS, n (%)	1423	(41.6)	663	(45.2)	760	(39.0)	<0.001
Number of MetS components in subjects with MetS	3.60	±0.73	3.63	±0.75	3.54	±0.70	0.002
BP ≥ 130/85 mmHg, n (%)	2561	(74.9)	998	(68.0)	1563	(80.2)	<0.001
FPG ≥ 100 mg/dL, n (%)	1294	(37.9)	502	(34.2)	793	(40.6)	<0.001
TGC ≥150 mg/dL, n (%)	989	(28.9)	354	(24.1)	635	(32.6)	<0.001
HDL-C mg/dL < 40 men, <50 mg/dL women, n (%)	973	(28.5)	546	(37.2)	427	(21.9)	<0.001
WC ≥ 88 cm women, ≥102 cm men, n (%)	1966	(57.5)	1018	(69.3)	948	(48.6)	<0.001

The values are presented as means ± standard deviations for continuous data and as numbers and proportions for categorical data. MD, Mediterranean diet; SBP, systolic blood pressure; DBP, diastolic blood pressure; LDL, low–density lipoprotein; HDL, high–density lipoprotein; FPG, fasting plasma glucose; HbA1c, glycosylated hemoglobin; BMI, body mass index; WC, waist circumference; MetS, metabolic syndrome; BP, blood pressure; TGC, triglycerides. The *p* value: differences between men and women.

**Table 3 nutrients-16-01948-t003:** Pearson correlation between Mediterranean diet and components of metabolic syndrome globally and by sex.

MD (Total Score)	Global	Women	Men
Number of MetS components	−0.196 **	−0.220 **	−0.185 **
SBP, (mmHg)	−0.122 **	−0.106 **	−0.106 **
DBP, (mmHg)	−0.172 **	−0.123 **	−0.184 **
FPG, (mg/dL)	−0.118 **	−0.140 **	−0.097 **
Triglycerides, (mg/dL)	−0.157 **	−0.149 **	−0.149 **
HDL cholesterol, (mg/dL)	0.171 **	0.179 **	0.130 **
WC, cm	−0.174 **	−0.198 **	−0.116 **

MD, Mediterranean diet; SBP, systolic blood pressure; DBP, diastolic blood pressure; HDL, high–density lipoprotein; FPG, fasting plasma glucose; WC, waist circumference; MetS, metabolic syndrome. Pearson coefficient. ** *p* < 0.001.

**Table 4 nutrients-16-01948-t004:** Association of the Mediterranean diet with the number and components of the metabolic syndrome globally and by sex using multiple regression analysis.

Global	β	(IC	95%)	*p*
Number of components MetS	−0.336	(−0.393	to −0.280)	<0.001
SBP, (mmHg)	−0.011	(−0.015	to −0.008)	<0.001
DBP, (mmHg)	−0.029	(−0.035	to −0.022)	<0.001
FPG, (mg/dL)	−0.009	(−0.012	to −0.006)	<0.001
Triglycerides, (mg/dL)	−0.004	(−0.004	to −0.003)	<0.001
HDL cholesterol, (mg/dL)	0.023	(0.018	to 0.028)	<0.001
WC, cm	−0.026	(−0.032	to −0.020)	<0.001
**Women**				
Number of components MetS	−0.314	(−0.396	to −0.231)	<0.001
SBP, (mmHg)	−0.009	(−0.014	to −0.004)	<0.001
DBP, (mmHg)	−0.021	(−0.030	to −0.012)	<0.001
FPG, (mg/dL)	−0.006	(−0.010	to −0.002)	0.002
Triglycerides, (mg/dL)	−0.004	(−0.005	to −0.002)	<0.001
HDL cholesterol, (mg/dL)	0.021	(0.014	to 0.027)	<0.001
WC, cm	−0.028	(−0.037	to −0.020)	<0.001
**Men**				
Number of components MetS	−0.349	(−0.427	to −0.271)	<0.001
SBP, (mmHg)	−0.013	(−0.018	to −0.008)	<0.001
DBP, (mmHg)	−0.034	(−0.043	to −0.026)	<0.001
FPG, (mg/dL)	−0.011	(−0.014	to −0.007)	<0.001
Triglycerides, (mg/dL)	−0.003	(−0.004	to −0.002)	<0.001
HDL cholesterol, (mg/dL)	0.021	(0.014	to 0.029)	<0.001
WC, cm	−0.023	−(0.032	to −0.014)	<0.001

Multiple regression analysis using number of MetS components, SBP, DBP, FPG, triglycerides, HDL cholesterol, and WC as dependent variables, and as independent variables were Mediterranean diet score, and as adjustment variables were age, sex, and antihypertensive drugs, hypoglycemic and lipid-lowering agents. MetS, metabolic syndrome; MD, Mediterranean diet; SBP, systolic blood pressure; DBP, diastolic blood pressure; HDL, high–density lipoprotein; FPG, fasting plasma glucose; WC, waist circumference.

**Table 5 nutrients-16-01948-t005:** Association of the adherence to the MD with metabolic syndrome and its components globally and by sex logistic regression analysis.

Global	OR	(95%CI)		*p*
MetS	0.555	(0.474	to 0.650)	<0.001
BP ≥ 130/85 mmHg	0.634	(0.522	to 0.771)	<0.001
FPG ≥ 100 mg/dL	0.615	(0.511	to 0.740)	<0.001
TGC ≥ 150 mg/dL	0.641	(0.546	to 0.753)	<0.001
HDL-C mg/dL < 40 men, <50 mg/dL women	1.700	(1.442	to 2.005)	<0.001
WC ≥ 88 cm women, ≥102 cm men	0.741	(0.639	to 0.859)	<0.001
**Women**				
MetS	0.478	(0.375	to 0.611)	<0.001
BP ≥ 130/85 mmHg	0.656	(0.489	to 0.879)	0.005
FPG ≥ 100 mg/dL	0.584	(0.437	to 0.781)	<0.001
TGC ≥ 150 mg/dL	0.770	(0.597	to 0.994)	0.045
HDL-C mg/dL < 40 men, <50 mg/dL women	1.715	(1.369	to 2.149)	<0.001
WC ≥ 88 cm women, ≥102 cm men	0.698	(0.551	to 0.884)	0.003
**Men**				
MetS	0.637	(0.517	to 0.785)	<0.001
BP ≥ 130/85 mmHg	0.624	(0.478	to 0.814)	0.001
FPG ≥ 100 mg/dL	0.640	(0.503	to 0.816)	<0.001
TGC ≥ 150 mg/dL	0.607	(0.491	to 0.751)	0.001
HDL-C mg/dL < 40 men, <50 mg/dL women	1.664	(1.304	to 2.122)	<0.001
WC ≥88 cm women, ≥102 cm men	0.783	(0.647	to 0.948)	0.012

Logistic regression analysis using MetS and its components as dependent variables, Mediterranean diet score ≥ 7 as independent variables, and as adjustment variables were age, sex, and consumption of antihypertensive drugs, hypoglycemic and lipid-lowering agents. MD, Mediterranean diet; MetS, metabolic syndrome; BP, blood pressure ≥ 130/85 mmHg; FPG, fasting plasma glucose ≥ 100 mg/dL; TGC, triglycerides ≥ 150 mg/dL; HDL-C, high–density lipoprotein < 40 men mg/dL, <50 mg/dL women; WC, waist circumference ≥ 88 cm women, ≥102 cm men.

## Data Availability

The variables used in the analyses carried out to obtain the results of this work are available upon reasonable request to the corresponding author.

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
