# Peer review of "Relationship between the Mediterranean Diet and Metabolic Syndrome and Each of the Components That Form It in Caucasian Subjects: A Cross-Sectional Trial"

_nutrients, 2024, doi:10.3390/nu16121948_

Round 1
Reviewer 1 Report
Comments and Suggestions for Authors
The manuscript by Gómez-Sánchez et al entitled “Relationship of Mediterranean Diet with Metabolic Syndrome and its components in Caucasian subjects. A cross‑sectional trial” had as its primary objective, to study the relationship of the Mediterranean diet (MD) with metabolic syndrome (MetS) and its components in Caucasian subjects and, secondly, to evaluate the presence of sex-related differences.
The study is very interesting, especially considering the health and socio-economic problems linked to metabolic syndrome and its causes.
There are some aspects that I believe deserve more attention from the authors.
1) L31: 60.14±9.14 years. SD or SE?
2) L83-85: authors should cite at least some of these studies and report key findings. In general, I find that the introduction section is quite poor in background information.
3) L105-106: I think the authors could report this information, at least briefly
4) L199: “percentage was 36.8% (women 40.3%, men 34.3).” How can these two facts be reconciled? on the one hand, it seems that men reach higher MDS, on the other, it is said that women have greater adherence to the Mediterranean diet
5) L199-201: “The percentage of subjects with MetS 199 was 41.6% (women 45.2%, men 39.0%) and the mean number of MetS components was 200 2.28±1.33 (women 2.33±1.44, men 2.24±1.25).” also in this case something is not right: the authors say that they are talking about metabolic syndrome if the individual has at least 3 of the 5 parameters considered. How can the average be less than 3?
6) Figure 2: I believe there is an error in the distribution of women: it is perfectly identical to the total one (figure 1)
7) Figure 3: where are the asterisks? what do the colors indicate?
8) L229-232: are they significant?
9) Table 3: I have doubts that the significances are all the same
10) L271-272: I think this sentence needs to be rewritten: the two statements are not in opposition.
11) Discussion: in general the authors limit themselves to reporting what others have found (which should go in the introduction) and do not discuss their results in light of their objectives.
12) L339-343: other factors may be involved. A recent study, using artificial intelligence analysis, related adherence to the Mediterranean diet, BP, HDL levels, triglycerides, FBG and metabolic syndrome in males and females to their olfactory function. This is not an aspect to be underestimated given the role that smell plays in eating behavior and food choice (https://doi.org/10.3390/metabo13020206).
The conclusions are weak and lack considerations on the second objective of the study.
Author Response
Y

Reviewer 2 Report
Comments and Suggestions for Authors
The manuscript “Relationship of Mediterranean Diet with Metabolic Syndrome and its components in Caucasian subjects” is well written and discussed. However, some aspects should be improved:
- Please insert more works and description in the introduction about the way in which they were carried out (some description).
- In the material and methods (2.2. Study population), please include a Flowchart of patients included in the study.
- Please improve the conclusions.
For these reasons, I consider the manuscript accept with major revisions on Nutrients.
Author Response
y
